# Discrete Memristor and Discrete Memristive Systems

**DOI:** 10.3390/e24060786

**Published:** 2022-06-03

**Authors:** Shaobo He, Donglin Zhan, Huihai Wang, Kehui Sun, Yuexi Peng

**Affiliations:** 1School of Physics and Electronics, Central South University, Changsha 410083, China; heshaobo@csu.edu.cn (S.H.); zdl_csu540@163.com (D.Z.); kehui@csu.edu.cn (K.S.); 2School of Cyberspace Science, Xiangtan University, Xiangtan 411105, China; pyx244896301@163.com

**Keywords:** memristor, discrete modeling of memristor, fractional-order difference, digital circuits, memristive system

## Abstract

In this paper, we investigate the mathematical models of discrete memristors based on Caputo fractional difference and G–L fractional difference. Specifically, the integer-order discrete memristor is a special model of those two cases. The “*∞*”-type hysteresis loop curves are observed when input is the bipolar periodic signal. Meanwhile, numerical analysis results show that the area of hysteresis decreases with the increase of frequency of input signal and the decrease of derivative order. Moreover, the memory effect, characteristics and physical realization of the discrete memristors are discussed, and a discrete memristor with short memory effects is designed. Furthermore, discrete memristive systems are designed by introducing the fractional-order discrete memristor and integer-order discrete memristor to the Sine map. Chaos is found in the systems, and complexity of the systems is controlled by the parameter of the memristor. Finally, FPGA digital circuit implementation is carried out for the integer-order and fractional-order discrete memristor and discrete memristive systems, which shows the potential application value of the discrete memristor in the engineering application field.

## 1. Introduction

In 1971, Chua [1] suggested there should be a fourth electron component in addition to resistance, capacitance and inductance. This new component was proposed to be called a memristor. In 2008, researchers at Hewlett Packard Labs reported that the memristor could be realized using nanoscale material [2]. Since this, there has been explosive growth in the research into memristors. It should be noted that the concept of the generalized memristor and the three fingerprint characteristics [3] of memristors have been reported, which provides a guide for the design of memristors. Owing to their properties, including memory and intrinsic nonlinearity, memristors have many potential applications in fields such as flash memory [4], reconfigurable computing [5], logic circuits [6], synapses [7], neural networks [8,9], solving mazes [10] and oscillation circuits [11]. Among those, memristor-based nonlinear systems or circuits have rich dynamics such as multistability and chaos [12,13].

The physical implementation of the memristor is important for real applications of memristors in the future. However, to date, there have been almost no memristor devices on sale, in comparison with resistance, capacitance and inductance ones. At present, researchers of material science are trying to realize memristors based on different kinds of nanometer materials, such as ZnO [14], TaOx [15], ZnO-rGO [16], VO${}_{2}$ [17] and CuSO${}_{4}$ [18]. A nanoscale memristor device can be applied in the neuromorphic systems as the synapse [19]. It should be noted that those nanoscale memristor devices satisfy the definition of a memristor, but do not have good mathematical models for theoretical analysis. In fact, researchers have used alternative techniques to design memristor devices based on mathematical models. Analog circuit implementation of memristors is important, and there are many different implementations [20,21,22,23]. Meanwhile, FPGA implementation has aroused much interest among researchers due to its easily programmable, reconfigurable, controllable, precise and better performance [24,25]. To improve the theoretical framework and applications of memristors, it is necessary to investigate functional memristor emulators and their intrinsic features.

There are two main approaches to building mathematical models for real systems. One is to build continuous models, and the other is to establish discrete models. Differential calculus is used in continuous models, and difference calculus is used in discrete models. As a result, some practical methods are proposed based on the difference [26,27,28,29]. For instance, in digital image processing [26], image enhancement operators can be obtained using difference and fractional-order difference. Meanwhile, since we have continuous memristors, discrete memristors should also exist. In fact, the concept of discrete memristors was first mentioned by Wang et al. [30], but the proposed model is not satisfying. In 2020, He et al. [31] proposed a mathematical model for a discrete memristor based on difference. Later, they introduced a discrete memristor to a Hénon map [32] and a higher-dimensional chaotic map [33]. Meanwhile, Bao et al. [34,35,36,37] designed discrete memristor chaotic maps and investigated their multistability. Moreover, Simulink simulation of a discrete memristor has been carried out [38]. To date, most of the work has been carried out based on integer-order discrete memristors. To build mathematical models and explain the physical significance of discrete memristors, some issues should be noted:The mathematical model for integer-order discrete memristors is clear, but the physical significance of discrete memristors still needs to be explored.Since there are different kinds of fractional-order difference, it is necessary to build fractional-order discrete memristors using different differences. At present, the two widely used fractional-order differences are Caputo-like difference [39] and G–L difference [40].The characteristics of discrete memristors, such as the memory effect and frequency domain, should be discussed.Physical implementation of the designed discrete memristors should be carried out and discussed.

Figure 1 shows a short history of the study of memristors and the position of this paper. It was started by Chua in 1971, when the model for the generalized memristor and the three fingerprints for memristors were proposed. This indicated the rules for designing memristors. To date, the design of memristors has been investigated by scientists from different research fields, such as mathematical modeling, circuit implementation, nanoscale materials, and applications. Therefore, what is the role of discrete memristors in the memristor family? In our opinion, the research of discrete memristors and discrete memristive systems in this paper follows the dashed lines in Figure 1. We use Caputo difference and G–L difference to build the mathematical models of fractional-order discrete memristors and realize them in digital circuits. This shows that a discrete memristor can also be a device for real applications.

The outline of this paper is given as follows. In Section 2, definitions of fractional-order differences and generalized memristors are presented. In Section 3, models of fractional-order discrete memristors are proposed, numerical simulations are carried out, and the short-term memory effect is discussed. In Section 4, physical significance of the discrete memristor is discussed. In Section 5, two kinds discrete memristive systems are designed, and their dynamics are analyzed. In Section 6, FPGA implementations of discrete memristive systems are carried out. Finally, we summarize the whole analysis.

## 2. Preliminaries

### 2.1. Fractional-Order Differences

In this section, two different fractional-order differences are presented.

**Definition** **1**([39,42]). *For a given fractional-order α>0, α∉N, when $u\left(t\right)$ is defined in $N_{t_{0}}$, its Caputo difference is defined by*
(1)$$\begin{array}{c} {}^{C}\Delta_{t_{0}}^{\alpha }u\left(t\right):=\Delta_{t_{0}}^{-(m-\alpha )}\Delta^{m}u\left(t\right)\\ \;\;\;\;=\frac{1}{\Gamma (m-\alpha )}\sum_{s=t_{0}}^{t-(m-\alpha )}{\left(t-\sigma (s)\right)}^{\left(m-\alpha -1\right)}\Delta^{m}u\left(s\right) \end{array},$$
*where Γ(·) is the gamma function, $t\in N_{t_{0}+m-\alpha }$, $N_{t_{0}}:=\left\{t_{0},t_{0}+1,t_{0}+2,\cdots \right\}$, $m=\left\lceil \alpha \right\rceil$, and σ(s) denotes the next point in the time scale after s, namely $\sigma \left(s\right)=s+1$ for $s\in N_{t_{0}}$.*

Obviously, when α=1, the fractional difference becomes $\Delta u\left(t_{n}\right)=u\left(t_{n+1}\right)-u\left(t_{n}\right)$.

**Theorem** **1**([39,42]). *For the Caputo-like fractional-order system*
(2)$${}^{C}\Delta_{t_{0}}^{\alpha }u\left(t\right)=g\left(t+\alpha -1,u\left(t+\alpha -1\right)\right),$$
*its solution is given by*
(3)$$\begin{array}{c} u\left(t\right)=\sum_{k=0}^{m-1}\frac{{\left(t-t_{0}\right)}^{\left(k\right)}}{k!}\Delta^{k}u\left(t_{0}\right)\\ \;\;\;+\frac{1}{\Gamma (\alpha )}\sum_{s=t_{0}+m-\alpha }^{t-\alpha }\frac{g\left(s+\alpha -1,u(s+\alpha -1)\right)}{{\left(t-\sigma (s)\right)}^{1-\alpha }} \end{array},$$
*where u(t) is the system variable, g(·) is the system equation and $\Delta^{k}u\left(t_{0}\right)=u_{k}$, k = 0, 1, ⋯, m−1, $m=\left\lceil \alpha \right\rceil$.*

**Definition** **2**( [40]). *The G–L fractional-order difference is defined by*
(4)$${}^{G}\Delta_{t_{0}}^{\alpha }g\left(t_{n}\right)=\sum_{j=0}^{n}{\left(-1\right)}^{j}\left(\begin{array}{c} \alpha \\ j \end{array}\right)g\left(t_{n-j}\right),$$
*where α is the fractional difference order, and $\left(\begin{array}{c} \alpha \\ j \end{array}\right)$ is denoted as*
(5)$$\left(\begin{array}{c} \alpha \\ j \end{array}\right)=\frac{\Gamma (\alpha +1)}{\Gamma (j+1)\Gamma (\alpha -j+1)},$$
*and $\left(\begin{array}{c} \alpha \\ 0 \end{array}\right)=1$.*

When α=1, we have
(6)$$\begin{array}{c} {}^{G}\Delta_{t_{0}}^{\alpha }x\left(t_{n}\right)=x\left(t_{n}\right)-x\left(t_{n-1}\right) \end{array}.$$
It can be seen that is a forward-difference operator.

**Definition** **3**([43]). *For a given G–L fractional-order discrete system with initial condition $x(t_{0})$ defined by*
(7)$${}^{G}\Delta_{t_{0}}^{\alpha }x\left(t_{n}\right)=g\left(x\left(t_{n-1}\right),t_{n-1}\right),$$
*since*
(8)$$\begin{array}{c} {}^{G}\Delta_{t_{0}}^{\alpha }x\left(t_{n}\right)=\sum_{j=0}^{n}{\left(-1\right)}^{j}\left(\begin{array}{c} \alpha \\ j \end{array}\right)x\left(t_{n-j}\right)\\ =x\left(t_{n}\right)+{\left(-1\right)}^{1}\left(\begin{array}{c} \alpha \\ 1 \end{array}\right)x\left(t_{n-1}\right)\\ +{\left(-1\right)}^{2}\left(\begin{array}{c} \alpha \\ 2 \end{array}\right)x\left(t_{n-2}\right)+\cdots +{\left(-1\right)}^{n}\left(\begin{array}{c} \alpha \\ n \end{array}\right)x\left(t_{0}\right)\\ =x\left(t_{n}\right)+\sum_{j=1}^{n}G_{j}^{\left(\alpha \right)}x\left(t_{n-j}\right) \end{array},$$
*the solution of this system is denoted as*
(9)$$x\left(t_{n}\right)=g\left(x\left(t_{n-1}\right),t_{n-1}\right)-\sum_{j=1}^{n}G_{j}^{\left(\alpha \right)}x\left(t_{n-j-1}\right),$$
*By formula (Equation 5)*
(10)$$G_{j}^{\alpha }={\left(-1\right)}^{j}\frac{\Gamma (\alpha +1)}{\Gamma (j+1)\Gamma (\alpha -j+1)}$$
*and $G_{0}^{\alpha }=1$.*

In Definition 3, $G_{j}^{\left(\alpha \right)}$ is the coefficient of the history data, with $G_{j}^{\left(\alpha \right)}=0$ (*j* = −2, −1) and $G_{j}^{\left(\alpha \right)}={\left(-1\right)}^{j}\left(\begin{array}{c} \alpha \\ j \end{array}\right)$, (*j* = 0, 1, 2, ⋯).

### 2.2. The Generalized Memristor

The definition of the generalized memristor is presented in Definition 3. It shows the relationship between the input i(t) and output y(t). For example, if i(t) and y(t) correspond to current and voltage of two circuit variables, respectively, it defines a current-controlled memristive system or a charge-controlled memristor.

**Definition** **4**([41]). *The generalized continuous memristor is defined by*
(11)$$\left\{\begin{array}{c} y\left(t\right)=g\left(x\left(t\right),i\left(t\right),t\right)i\left(t\right)\\ \dot{x}\left(t\right)=f\left(x\left(t\right),i\left(t\right),t\right) \end{array}\right.,$$
*where x(t) is the internal state variable of the memristor, g(·) is a function that represents the value of the memristor, and f(·) is a continuous function.*

**Definition** **5.**
*Describes the case of a continuous memristor. Here, we give a definition for the generalized discrete memristor.*


**Definition** **6.**
*The generalized fractional discrete memristor is defined as*

(12)
$$\left\{\begin{array}{c} y\left(t_{n}\right)=g\left(x_{n},i\left(t_{n}\right),t_{n}\right)i\left(t_{n}\right)\\ \Delta_{t_{0}}^{\alpha }x\left(t_{n}\right)=f\left(x\left(t_{n-1}\right),i\left(t_{n-1}\right),t_{n-1}\right) \end{array}\right.,$$

*where g(·) is a function that represents the value of the discrete memristor, and f(·) is a continuous function. As with the continuous memristor, the generalized discrete memristor also describes the relationship between the input signal and the output signal.*


The input and output signal are not limited by the voltage and current, but can be any discrete signals.

According to Definition 4, a universal fractional-order discrete memristor is defined by
(13)$$\left\{\begin{array}{c} y\left(t_{n}\right)=g\left(x\left(t_{n}\right)\right)i\left(t_{n}\right)\\ \Delta_{t_{0}}^{\alpha }x\left(t_{n}\right)=f\left(x\left(t_{n-1}\right),i\left(t_{n-1}\right),t_{n-1}\right) \end{array}\right..$$
An ideal fractional-order discrete universal memristor is defined by
(14)$$\left\{\begin{array}{c} y\left(t_{n}\right)=g\left(x\left(t_{n}\right)\right)i\left(t_{n}\right)\\ \Delta_{t_{0}}^{\alpha }x\left(t_{n}\right)=f\left(x\left(t_{n-1}\right)\right)i\left(t_{n-1}\right) \end{array}\right..$$
In this paper, the ideal fractional-order discrete memristor is defined as
(15)$$\left\{\begin{array}{c} y\left(t_{n}\right)=g\left(x\left(t_{n}\right)\right)i\left(t_{n}\right)\\ \Delta_{t_{0}}^{\alpha }x\left(t_{n}\right)=f\left(i\left(t_{n-1}\right)\right) \end{array}\right..$$

Here, g(·) is defined in two cases:Case 1:$g\left(x\right)=a_{1}+b_{1}x$.Case 2:$g\left(x\right)=a_{2}+b_{2}\left|x\right|$.

In this paper, we fix $a_{1}=0.8$, $b_{1}=-0.0001$, $a_{2}=0.667$, $b_{2}=1.4828\times 10^{-4}$, and the periodic input signals are defined as
(16)$$i\left(t_{n}\right)=A\sin\left(\omega t_{n}\right),$$
where $t_{n}$ = 0, 1, 2, ⋯.

**Theorem** **2**([3]). *The three fingerprints determine whether a device is a memristor or not. The fingerprints are:*
(1)*When it is driven by a bipolar periodic signal, the device must exhibit a “pinched hysteresis loop” in the voltage–current plane, assuming the response is periodic.*(2)*Starting from some critical frequency, the hysteresis loop area should decrease monotonically as excitation frequency increases.*(3)*The pinched hysteresis loop should shrink to a single-valued function when the frequency tends to infinity.*

For the discrete memristor, it should have the three fingerprints when the input of the discrete memristor contains bipolar periodic signals. For the function f(·), there are many different choices, including piecewise nonlinearity, linear nonlinearity, absolute value, linear function, sine function, and square function [44]. In this manuscript, we use the idea of memristor models including HP memristors (linear function) and absolute function, to show the effectiveness of the proposed methods. However, for function *g*(·), this does not offer much choice, and they are $\dot{x}\left(t\right)=ki\left(t\right)$ and $\Delta x\left(t_{n}\right)=ki\left(t_{n}\right)$ for the continuous case and discrete case, respectively, where *k* is strength. In this paper, we also use three fingerprints to verify the rationality of the designed discrete memristors.

## 3. Models of Discrete Memristor

### 3.1. Fractional-Order G–L Difference-Based Model

Based on the G–L definition, the fractional-order discrete memristor is defined by
(17)$$\left\{\begin{array}{c} \psi \left(t_{n}\right)=F\left(\varphi ,i\left(t_{n}\right),t_{n}\right)i\left(t_{n}\right)\\ {}^{G}\Delta_{t_{0}}^{\alpha }\varphi \left(t_{n}\right)=f\left(\varphi \left(t_{n-1}\right),i\left(t_{n-1}\right),t_{n-1}\right) \end{array}\right..$$

According to Definition 4, the internal valuable in the G–L fractional-order discrete memristor is defined by
(18)$$\varphi \left(t_{n}\right)=f\left(\varphi \left(t_{n-1}\right),i\left(t_{n-1}\right),t_{n-1}\right)-\sum_{j=1}^{n}G_{j}^{\left(\alpha \right)}\varphi \left(t_{n-j}\right).$$
As a result, this discrete memristor can be written as
(19)$$\left\{\begin{array}{c} \psi \left(t_{n}\right)=F\left(\varphi \left(t_{n}\right),i\left(t_{n}\right),t_{n}\right)i\left(t_{n}\right)\\ \begin{array}{c} \varphi \left(t_{n}\right)=f\left(\varphi \left(t_{n-1}\right),i\left(t_{n-1}\right),t_{n-1}\right)\\ \;\;\;-\sum_{j=1}^{n}G_{j}^{\left(\alpha \right)}\varphi \left(t_{n-1}\right) \end{array} \end{array}\right..$$

To simulate the system, we need to calculate coefficient $G_{j}^{\left(\alpha \right)}$ in a more effective way.

First, we have the following equation
(20)$$\Gamma \left(\alpha +1\right)=\alpha \Gamma \left(\alpha \right).$$

According to the definition of the coefficient, *j* = (1, 2, 3, ⋯), we have
(21)$$G_{j-1}^{\left(\alpha \right)}={\left(-1\right)}^{j-1}\frac{\Gamma (\alpha +1)}{\Gamma (j)\Gamma (\alpha -j+2)}.$$

Then,
(22)$$\begin{array}{c} G_{j}^{\left(\alpha \right)}={\left(-1\right)}^{j}\frac{\Gamma (\alpha +1)}{\Gamma (j+1)\Gamma (\alpha -j+1)}\\ \;\;\;\;=-{\left(-1\right)}^{j-1}\frac{\Gamma (\alpha +1)}{\Gamma (j)\Gamma (\alpha -j+2)}\cdot \frac{\Gamma (j)\Gamma (\alpha -j+2)}{\Gamma (j+1)\Gamma (\alpha -j+1)}\\ \;\;\;\;=-G_{j-1}^{\left(\alpha \right)}\frac{\Gamma (j)\Gamma (\alpha -j+2)}{\Gamma (j+1)\Gamma (\alpha -j+1)}\\ \;\;\;\;=-G_{j-1}^{\left(\alpha \right)}\frac{\Gamma \left(j\right)\left(\alpha -j+1\right)\Gamma \left(\alpha -j+1\right)}{j\Gamma (j)\Gamma (\alpha -j+1)}\\ \;\;\;\;=-\frac{\alpha -j+1}{j}G_{j-1}^{\left(\alpha \right)}=\left(1-\frac{\alpha +1}{j}\right)G_{j-1}^{\left(\alpha \right)} \end{array}.$$

When j=0, $G_{0}^{\left(\alpha \right)}=1$. Thus, we can obtain the values of $G_{j}^{\left(\alpha \right)}$ using the relationship as given in Equation (Equation 22).

An example of the G–L fractional-order discrete memristor is given by
(23)$$\left\{\begin{array}{c} y\left(t_{n}\right)=g\left(x\left(t_{n}\right)\right)i\left(t_{n}\right)\\ {}^{G}\Delta_{t_{0}}^{\alpha }x\left(t_{n}\right)=ki\left(t_{n-1}\right) \end{array}\right..$$

Thus, the model is denoted as
(24)$$\left\{\begin{array}{c} y\left(t_{n}\right)=g\left(x\left(t_{n}\right)\right)i\left(t_{n}\right)\\ x\left(t_{n}\right)=i\left(t_{n-1}\right)-\sum_{j=1}^{n}G_{j}^{\left(\alpha \right)}x\left(t_{n-j}\right) \end{array}\right..$$

The hysteresis loop of the G–L difference-based discrete memristor with different functions g(·), order α and frequency ω is illustrated in Figure 2, where Figure 2a,b are plotted using A=4, ω=0.001 and different order α and Figure 2c,d are plotted using A=4, α=0.95 and different frequency ω. Obviously, the “*∞*” pinched hysteresis loops are observed with the given bipolar periodic inputs. As with the Caputo-like difference-based memristor, the area in the hysteresis loop shrinks with the decrease in derivative-order α and increase in frequency ω. According to the numerical simulation analysis results, the designed fractional-order memristor satisfies the three fingerprints of the memristor.

### 3.2. Fractional-Order Caputo Difference-Based Model

Based on Caputo-like difference, the mathematical model of the fractional-order discrete memristor is denoted as [31]
(25)$$\left\{\begin{array}{c} \psi \left(t_{n}\right)=g\left(\varphi ,u,t_{n}\right)i\left(t_{n}\right)\\ {}^{C}\Delta_{t_{0}}^{\alpha }\varphi \left(t\right)=f\left(\varphi ,i,t+\alpha -1\right) \end{array}\right.,$$
where g(·) and f(·) are the nonlinear functions and φ is the internal state variable of the memristor (equivalent to the “charge” or “magnetic flux” in the continuous memristors).

According to Definition 2, the solution of the second equation in Equation (Equation 25) is given by
(26)$$\varphi \left(t_{n}\right)=\varphi \left(t_{0}\right)+\frac{1}{\Gamma \left(\alpha \right)}\sum_{j=1}^{n}C_{j}^{\left(\alpha \right)}f\left(\varphi \left(t_{j-1}\right),u,t_{j-1}\right),$$
where the coefficient is defined by
(27)$$C_{j}^{\left(\alpha \right)}=\frac{\Gamma \left(n-j+\alpha \right)}{\Gamma \left(n-j+1\right)},$$
and it can be calculated in a more effective way, which is denoted as
(28)$$C_{j}^{\left(\alpha \right)}=\left\{\begin{array}{c} \Gamma \left(\alpha \right),j=0\\ C_{j-1}^{\left(\alpha \right)}\frac{\alpha +j-1}{j},j=1,2,\cdots \end{array}\right..$$

Finally, the fractional-order discrete memristor is rewritten as
(29)$$\left\{\begin{array}{c} \psi \left(t_{n}\right)=g\left(\varphi ,u,t_{n}\right)u\left(t_{n}\right)\\ \varphi \left(t_{n}\right)=\varphi \left(t_{0}\right)+\sum_{j=1}^{n}C_{j}^{\left(\alpha \right)}f\left(\varphi \left(t_{j-1}\right),u,t_{j-1}\right) \end{array}\right..$$

The nonlinear functions including g(·) and f(·) in the memristor have different kinds of considerations.

An example of the Caputo fractional-order discrete memristor is given by
(30)$$\left\{\begin{array}{c} y\left(t_{n}\right)=g\left(x\left(t_{n}\right)\right)i\left(t_{n}\right)\\ {}^{C}\Delta_{t_{0}}^{\alpha }x\left(t\right)=ki\left(t+\alpha -1\right) \end{array}\right..$$

Thus, the model with coefficient is
(31)$$\left\{\begin{array}{c} y\left(t_{n}\right)=g\left(x\left(t_{n}\right)\right)i\left(t_{n}\right)\\ x\left(t_{n}\right)=x\left(t_{0}\right)+\frac{k}{\Gamma \left(\alpha \right)}\sum_{j=1}^{n}C_{j}^{\left(\alpha \right)}i\left(t_{j-1}\right) \end{array}\right..$$

The voltage–current curves of the discrete memristor with different functions g(·), derivative-order α, and frequency ω are shown in Figure 3. Figure 3a,b are plotted using A=4, ω=0.001 and different derivative-order α. Figure 3c,d are plotted using A=4, α=0.9 and different frequency ω. This shows that there exist pinched hysteresis loops when the input signal is bipolarly periodic. The area of the hysteresis loop decreases with the decrease of derivative-order α and the increase of frequency ω. First, this shows that the fractional difference makes the discrete memristor have a lower frequency domain, since the area of the hysteresis loops decreases with order α. Second, this shows that the pinched hysteresis loop shrinks with frequency ω and becomes a line when ω=0.5. It can be seen that the designed fractional-order memristor satisfies the three fingerprints of the memristor.

### 3.3. Integer-Order Discrete Memristor

For the G–L fractional-order difference-based discrete memristor, when α=1, the coefficients are given by
(32)$$G_{0}^{\left(1\right)}=1,G_{1}^{\left(1\right)}=-1,G_{j}^{\left(1\right)}=0\left(j=2,3,\cdots \right).$$
Thus, the “charge” or “magnetic flux” in the integer-order discrete memristor is denoted as
(33)$$\varphi \left(t_{n}\right)=\varphi \left(t_{n-1}\right)+f\left(\varphi \left(t_{n-1}\right),i\left(t_{n-1}\right),t_{n-1}\right).$$
For the Caputo difference-based fractional-order discrete memristor, the coefficients are $G_{j}^{\left(1\right)}=1\left(j=0,1,2,\cdots \right)$ for α=1. The solution of the “charge” or “magnetic flux” is denoted as
(34)$$\begin{array}{c} \varphi \left(t_{n}\right)=\varphi \left(t_{0}\right)+\sum_{j=1}^{n}f\left(\varphi \left(t_{j-1}\right),i\left(t_{j-1}\right),t_{j-1}\right)\\ \;\;\;\;\;\;=\varphi \left(t_{0}\right)+\sum_{j=1}^{n-1}f\left(\varphi \left(t_{j-1}\right),i\left(t_{j-1}\right),t_{j-1}\right)\\ \;\;\;\;\;\;\;\;+f\left(\varphi \left(t_{n-1}\right),i\left(t_{n-1}\right),t_{n-1}\right)\\ \;\;\;\;\;\;=\varphi \left(t_{n-1}\right)+f\left(\varphi \left(t_{n-1}\right),i\left(t_{n-1}\right),t_{n-1}\right) \end{array}.$$
Thus, for both the Caputo difference-based memristor and the G–L difference-based memristor, when α=1, the integer-order counterpart is defined by
(35)$$\left\{\begin{array}{c} \psi \left(t_{n}\right)=g\left(\varphi_{n},i\left(t_{n}\right),t_{n}\right)i\left(t_{n}\right)\\ \varphi \left(t_{n}\right)=\varphi \left(t_{n-1}\right)+f\left(\varphi \left(t_{n-1}\right),i\left(t_{n-1}\right),t_{n-1}\right) \end{array}\right..$$

It should be noted that the integer-order discrete memristor is a special case of fractional-order discrete memristors. At present, integer-order discrete memristors are being investigated. For example, there is a chaotic map using a second-order discrete memristor [34], which is defined by
(36)$$\left\{\begin{array}{c} x_{n+1}=x_{n}-kx_{n}\cos\varphi_{n}\\ \varphi_{n+1}=\varphi_{n}+\varepsilon x_{n} \end{array}\right.,$$
where the discrete memristor is given by
(37)$$\left\{\begin{array}{c} x_{n+1}=kx_{n}\cos\varphi_{n}\\ \varphi_{n+1}=\varphi_{n}+\varepsilon x_{n} \end{array}\right..$$
It is not difficult to verify that there is an “*∞*” hysteresis loop, which means that there exists a memristor in the given discrete memristor.

### 3.4. Short-Term Memory Effects and Frequency Domain

#### 3.4.1. The Imperfect Memory Effect

As shown in Figure 4a, the integer-order discrete memristor has the ideal memory effect, since $\varphi \left(t_{n}\right)$ is the summation of all the history data. The reason for this simplified calculation is that all the coefficients for the history data are same, which is one.

However, when 0<α<1, then there is a different coefficient for each history data point, and the values of the coefficients based on the two fractional calculi are different. The kernel functions $C_{j}^{\left(\alpha \right)}$ and $G_{j}^{\left(\alpha \right)}$ are plotted in Figure 4, where n=100. As shown in Figure 4, for the Caputo difference, $C_{j}^{\left(\alpha \right)}$ tends to zero with a speed that is not as fast as the $G_{j}^{\left(\alpha \right)}$ from G–L difference. However, when α=1, *j* = 0, 1, 2, ⋯, and we have a horizontal line for $C_{j}^{\left(\alpha \right)}$. In addition, α=1, $C_{j}^{\left(\alpha \right)}=1$ and for *j* = 2, 3, 4, ⋯, there is no line for $G_{j}^{\left(1\right)}$, because when α<1, the coefficients for different positions are different. First, for fractional-order cases, the calculation cost increases with iterations and, in real applications, we need to balance calculation cost and accuracy. Second, when 0<α<1, the memory effect is not “perfect”. Thus, from the point view of this paper, there exists an “imperfect memory effect" in the fractional-order discrete memristors.

#### 3.4.2. Short-Term Memory Memristor

The short-term memory effect of the fractional-order difference has been discussed by different researchers [45,46]. It sacrifices the accuracy of the model to balance calculation cost. In this paper, we use the sliding-window memory scheme. It means that current data are related to the previously limited number of data. For the Caputo-like fractional-order discrete memristor with short-term memory effect, it is defined as
(38)$$\begin{array}{c} \varphi \left(t_{n}\right)=\varphi \left(t_{0}\right)+\\ \;\;\;\frac{1}{\Gamma \left(\alpha \right)}\sum_{j=1}^{L}C_{j}^{\left(\alpha \right)}f\left(\varphi \left(t_{n-j}\right),i\left(t_{n-j}\right),t_{n-j}\right) \end{array},$$
where $L=\min\left\{n,L_{C}\right\}$, and $L_{C}$ is the memory length. For the Caputo fractional-order discrete memristor given in Equation (Equation 30), the short-term memory model is given by
(39)$$x\left(t_{n}\right)=x\left(t_{0}\right)+\frac{k}{\Gamma \left(\alpha \right)}\sum_{j=1}^{L}C_{j}^{\left(\alpha \right)}i\left(t_{n-j}\right).$$

The bipolarly periodic input signals and parameters of the memristor function g(·) are given in Equation (Equation 16). Let α=0.95, $x(t_{0})=0.01$. The pinched hysteresis loops of the Caputo difference-based memristor with different memory length and frequency are shown in Figure 5. This shows that the area of the pinched hysteresis loops decreases with an increase of frequency and decrease of memory length. Thus, the memristor with short-term memory effect also satisfies the definition of the memristor, and the memory length should not be too short. For the given example, memory length should be larger than 1000.

For the G–L definition-based model, the memory part is in the second item. Here, the G–L definition-based fractional-order discrete memristor with memory length *L* is defined by
(40)$$\varphi \left(t_{n}\right)=f\left(\varphi \left(t_{n-1}\right),u,t_{n-1}\right)-\sum_{j=1}^{L}G_{j}^{\left(\alpha \right)}\varphi \left(t_{n-j}\right),$$
where $L=\min\left\{n,L_{GL}\right\}$. Although $G_{j}^{\left(\alpha \right)}$ decreases with *j* significantly as shown in Figure 4, it is still necessary to investigate how memory length affects the nonlinearity of the memristor. The discrete memristor given in Section 3.2 with short-term memory effect is defined by
(41)$$x\left(t_{n}\right)=i\left(t_{n-1}\right)-\sum_{j=1}^{L}G_{j}^{\left(\alpha \right)}x\left(t_{n-j}\right).$$

The bipolar periodic input signals and parameters of the memristor function g(·) are given in Equation (Equation 16). Let α=0.95, $x(t_{0})=0.01$. The pinched hysteresis loops of the G–L difference-based memristor with different memory lengths and frequencies are shown in Figure 6. This also shows that the area in the pinched hysteresis loops decreases with an increase in frequency and decrease in memory length. As shown in Figure 6, when the memory length is larger than 10, the nonlinearity of the memristor is acceptable. However, the larger memory means a larger area in the pinched hysteresis loops. Compared with the Caputo difference memristor, the G–L difference-based memristor has much less memory for the satisfying nonlinearity.

## 4. Physical Significance of Discrete Memristor

At present, scientists use different symbols to represent different memristors. Generally, a solid rectangle represents an integer-order memristor, while a solid triangle represents a fractional-order memristor. However, there are no recognized symbols for discrete memristors. Here, Table 1 is the summary of the symbols used for different kinds of memristors. For the discrete memristor, it can be realized in a digital circuit where parameters are set by the control signal. In fact, the symbol represents a black box, where the interior structure of different memristors is different with different implementation techniques.

Table 2 shows that characteristics of different kinds of memristors are different.

Both continuous and discrete memristors have memory effects. In particular, there exists a perfect memory effect in integer-order memristors, according to their mathematical models.The theoretical work of continuous memristors has been investigated systematically. However, there is little work on discrete cases.There is some research regarding nano-device implementation of both continuous and discrete memristors. However, an issue should be resolved. The implemented nano-devices have “*∞*” hysteresis loop but usually do not relate to a mathematical model, and the “*∞*” hysteresis loops are not elegant. We believe that discrete memristor models can prove a useful tool for memristor nano-devices.Although there are reports of the FPGA implementation of continuous memristors, analog circuit implementation of continuous memristors is the main technical means. However, discrete memristors are naturally supposed to be realized in digital circuits including DSP and FPGA.Continuous memristors can be used in continuous systems such as nonlinear chaotic systems and neural networks.

In real applications, there are many discrete systems, such as chaotic maps and discrete neural networks. Thus, how to introduce memristors to those discrete processes is a challenging topic. However, the proposed discrete memristors prove good models for these issues.

At present, there is no uniform definition for fractional calculus and difference. Therefore, the proposed models for fractional-order discrete memristors may not be final results. We believe there could be better models and implementations for discrete memristors. Here, we propose two different fractional-order discrete memristors using Caputo difference and G–L difference.

Moreover, it is necessary to discuss the scheme for the realization of discrete memristors in analog circuits. Here, a feasible way is presented in Figure 7, which is proposed by Wu et al. [47]. Since the value of the memristor is decided by charge or magnetic flux, we sample the current or voltage of the circuit, and then the charge or magnetic flux is estimated in the microprocessor based on sample data. Finally, the output signal of the microprocessor decides the value of the variable resistance, which can be a digital potentiometer. Figure 7 provides a scheme based on the analog–digital hybrid circuit, but it still needs to be further investigated by engineers.

## 5. Discrete Memristive Systems

### 5.1. Design of Discrete Memristive Map

The Sine map is defined by
(42)$$y_{n}=\varepsilon \sin\left(\pi y_{n-1}\right),$$
where ε is the system parameter. Fixed ε=3, phase diagram and time series of the Sine map are shown in Figure 8. In this section, the integer-order and fractional-order discrete memristors are introduced to this system.

The design of discrete memristive chaotic maps is a hot topic. To date, as for the applications of discrete memristors in nonlinear systems, we have already introduced discrete memristors into chaotic maps, including the Sine map [31] and Hénon map [32] for better performance. However, designing a fractional-order chaotic memristive map is still a challenge. The block diagrams, as shown in Figure 9, are the Sine map and the discrete memristor Sine map. For a discrete memristive map, the signal $y_{n-1}$ is the input of the discrete memristor and the output of the memristor $V_{n-1}$ is then sent to the Sine map.

### 5.2. Caputo Fractional-Order Sine Map

According to Figure 9b, the fractional-order discrete memristive Sine map with Caputo discrete memristor is defined by [31]
(43)$$\left\{\begin{array}{c} \begin{array}{c} y\left(t_{n}\right)=\varepsilon \sin\left(\pi V_{n-1}\right)\\ V_{n-1}=y\left(t_{n-1}\right)g\left(q\left(t_{n-1}\right)\right) \end{array}\\ {}^{C}\Delta_{t_{0}}^{\alpha }q\left(t\right)=ki\left(t+\alpha -1\right) \end{array}\right.,$$
where g(q)=0.667+b|q|. When we considered the short-term memory effect, the third equation in this model is solved by
(44)$$q\left(t_{n}\right)=q\left(t_{0}\right)+\;\frac{k}{\Gamma \left(\alpha \right)}\sum_{j=1}^{L}C_{j}^{\left(\alpha \right)}i\left(t_{n-j}\right),$$
where $L=\min\left\{n,L_{C}\right\}$, and $L_{C}$ is the memory length. In this study, $L_{C}=1000$. The initial conditions of the system are $q_{0}=0.2$ and $y_{0}=0.63$.

Fixing ε=3 and letting *b* take different values including $b=1.4828\times 10^{-1}$, $b=1.4828\times 10^{-2}$, $b=1.4828\times 10^{-3}$ and $b=1.4828\times 10^{-4}$, phase diagrams of the Caputo fractional-order memristive Sine map are shown in Figure 10. This shows that the distribution of the points in the phase plane $y_{n}-y_{n+1}$ becomes concentrated with the decrease of parameter *b*. Thus, the complexity of the system can be affected by memristor parameter *b*.

### 5.3. G–L Fractional-Order Sine Map

Here, based on the G–L difference, the G–L memristor Sine map is defined as
(45)$$\left\{\begin{array}{c} y\left(t_{n}\right)=\varepsilon \sin\left(\pi V_{n-1}\right)\\ V_{n-1}=y\left(t_{n-1}\right)g\left(q\left(t_{n-1}\right)\right)\\ {}^{G}\Delta_{t_{0}}^{\alpha }q\left(t_{n}\right)=ky\left(t_{n-1}\right) \end{array}\right.,$$
where g(q)=0.667+b|q|. The solution of the third equation in Equation (Equation 45) is given by
(46)$$q\left(t_{n}\right)=ky\left(t_{n-1}\right)-\sum_{j=1}^{L}G_{j}^{\left(\alpha \right)}q\left(t_{n-j}\right),$$
where $L=\min\left\{n,L_{C}\right\}$, and $L_{C}$ is the memory length. Here, $L_{C}=50$. By putting $q(t_{n})$ as the first item of Equation (Equation 45), the model of the system G–L difference-based Sine map is obtained.

Fix ε=3 and let *b* take different values. Then, phase diagrams of the system are shown in Figure 11. These show that the phase diagram becomes to a Sine-like curve with the decrease of memristor parameter *b*. As shown in Figure 11, when b=1.4828, the points of data fill the whole space. This assumes the highest complexity compared to other cases, and the complexity of the system decreases with the decrease of *b*.

### 5.4. Integer-Order Discrete Sine Map

For both fractional-order cases, when the difference order α=1, the discrete memristive Sine map is defined as
(47)$$\left\{\begin{array}{c} y_{n}=\varepsilon \sin\left(\pi y_{n-1}g\left(q_{n-1}\right)\right)\\ q_{n}=q_{n-1}+ky_{n-1} \end{array}\right.,$$
which is an integer-order discrete memristive system. Here, g(q)=0.667+b|q|. ε and *b* are the two parameters of this system.

Let ε=3 and *b* take different values. Numerical analysis results of the system are shown in Figure 12. Similar results are obtained from the system. When *b* takes larger values, the points in the phase plane are more evenly distributed. Moreover, $q_{n}$ and $y_{n}$ are plotted. $q_{n}$ is the accumulation of time series $y_{n}$, which fluctuates on both sides of zero. In conclusion, complexity of the integer-order memristive Sine map can be modulated by the memristor parameter.

### 5.5. Complexity Analysis

#### 5.5.1. Maximum LEs of the Integer-Order System

Vary ε from 0 to 10 with step size of 0.01 and vary the parameter *b* from 0 to 1.4828 with step size of 0.0148. Maximum Lyapunov exponents (LEs) analysis results are shown in Figure 13. This shows that the maximum LEs increase with system parameter ε and memristor parameter *b*, and there are many periodic windows found. Meanwhile, the value of the maximum LEs can reach more than 8, which is larger than most of the chaotic maps. For instance, when ε=3, the Lyapunov exponent of the Sine map is 1.6721, which is much smaller than those values. Thus, the complexity of the Sine map is improved with the introduction of a discrete memristor.

#### 5.5.2. SampEn and NetPE Complexity Analysis

Generally, the complexity of time series means the time series are closer to a random sequence. However, there are many different approaches to measure complexity. In this paper, we use SampEn [48] and NetPE (network permutation entropy) [49] to measure the complexity of the time series generated by different chaotic maps. NetPE measures complexity by building a network using the Bandt–Pompe patterns and their weights. SampEn measures complexity by estimating the probability of generating new patterns in the time series. The greater the probability of generating new patterns, the higher the complexity of the series.

The NetPE algorithm [49] is defined as
(48)$$NetPE\left(x^{N},d,error\right)=-\frac{1}{\log\left(N-d+1\right)}\sum_{i=1}^{N-d+1}P\left(i\right)\log\left(P\left(i\right)\right),$$
where $x^{N}$ represents the time series *x* with length *N*, *d* is the Bandt–Pompe dimension, and error is the tolerance of difference between Bandt–Pompe vectors. More details about this algorithm can be found in [49]. In this study, d=3, error=0.02 and the length of time series N= 50,000.

For the given sequence $\left\{x^{N}:x_{1},x_{2},\cdots ,x_{N}\right\}$, the definition of SampEn [48] is given by
(49)$$\left\{\begin{array}{c} SamEn\left(m,r,N\right)=\ln\Phi^{m}\left(r\right)-\ln\Phi^{m+1}\left(r\right)\\ \Phi^{m}\left(r\right)={\left(N-m+1\right)}^{-1}\sum_{i=1}^{N-m+1}C_{i}^{m}\left(r\right)\\ C_{i}^{m}\left(r\right)={\left(N-m\right)}^{-1}\sum_{j=1,j\neq i}^{N-m+1}d_{i,j}^{m}\\ d_{i,j}^{m}=\max_{p=1,2,\cdots ,m}(|u\left(i+p+1\right)-u_{0}\left(i\right)| \end{array}\right.,$$
where *m* is the phase space dimension, *r* is the similarity tolerance, *N* is the sequence length, and m(m≤N−2) is the non-negative integer. In this paper, r=0.15 and m=3. In addition, the length of time series for SampEn is 2500.

Let ε=3 and α=0.95. Complexity analysis results with *b* varying are shown in Figure 14. Since Sine does not have parameter *b*, there is a transverse line presented in this figure for comparison. Let b=1.4828×0.8 and α=0.95. The complexity analysis results with variation of ε are shown in Figure 15. This shows that memristive Sine maps have higher complexity than the Sine map, which means that the complexity of the original chaotic map can be improved with the introduction of discrete memristors.

Since g(q)=0.667+b|q|, when *b* takes larger values, the contribution of *q* to the system is obvious. For instance, when $b=1.4828\times 10^{-4}$, g(q)≈0.667. When b=1.4828, g(q) will reach 2.1498. This makes the system more complex. The main reason for this is that the internal nonlinearity of the discrete memristor can significantly improve the complexity of the system when *b* takes large values. In conclusion, the discrete memristor can improve complex behaviors of the chaotic map.

## 6. Implementations of the Discrete Memristive Systems

### 6.1. FPGA Digital Circuit Design

According to the Equation (Equation 39), the digital circuit of a Caputo fractional-order discrete memristor is shown in Figure 16. It shows Reg_It and Reg_Ct are the core components to implement the Caputo fractional-order discrete memristor, where the former is the queue to store input current $It_{n}$ and the latter is the array to store the Caputo differential operator. Additionally, *n* and len represent the discrete time and the memory length, respectively. The value of len is *n* only when len>n, otherwise it is the set memory length. When the number of stored values reaches $Reg\_It^{\prime }s$ maximum memory length len, the values in it move to their next address, and the new input current is stored in the lowest address [0]. The Caputo differential operator is generated by $C_{n}^{q}$ operator module. The calculation process of the module is defined in Equation (Equation 28), and its digital implementation is presented in Figure 17a, where n=0,1,2,3,⋯,Reg_Ct[0]=Γ(q) and *q* is the derivative order. In the Caputo differential calculation part of Figure 16, the values in Reg_It and Reg_Ct are multiplied in accordance with the rule of the sum of address being len. The multiplied values are accumulated and then combined with constants k,Γ(q) and $m_{0}$ to form $x(t_{n})$, where $m_{0}=x\left(t_{0}\right)$. $x(t_{n})$ is the input into the compute unit, $g(x\left(t_{n}\right))$ is the output, and then it multiplies by $It_{n}$ to obtain the voltage output $Vt_{n}$, which also is $y(t_{n})$.

According to Equation (Equation 40), the digital circuit of the G–L fractional-order discrete memristor is shown in Figure 18. In Figure 18, Reg_Xt is the queue to store $x(t_{n})$ and Reg_Gt is the array to store G–L differential operator, where $Reg\_Xt\left[0\right]=x\left(t_{0}\right)$. The definition of this module is given in Equation (Equation 22) and its digital implementation is presented in Figure 17b, where *n* and *q* have the same meanings as above. In the G–L differential calculation part of Figure 18, the values in Reg_Xt and Reg_Gt are multiplied in the same manner as above. The multiplied results are accumulated and combined with $It_{n}$ to obtain $x(t_{n})$. On the one hand, the value of $x(t_{n})$ is stored in Reg_Xt; on the other hand, the value of $x(t_{n})$ is transmitted to compute the unit part to obtain $g(x\left(t_{n}\right))$, which multiplies with $It_{n}$ to obtain the voltage output $Vt_{n}$, which also is $y(t_{n})$. Moreover, when the number of stored values of Reg_Xt reaches the maximum storage length len, it will also perform the same operation as Reg_It above.

The model for the discrete memristive Sine map is shown in Figure 19. The designed Caputo fractional-order IP core and G–L fractional-order IP core are employed. Since the integer-order memristor is a special case of fractional-order memristor, we designed the IP core of the integer-order memristor directly in the memristive chaotic map, and have not presented the details. There is a selector, thus one of the three memristors works for the system and the proposed scheme can generate three different kinds of chaotic system.

To compare our models with Matlab simulation results, we designed a memristor chaotic test system that contains multiple types of discrete memristors and discrete memristive chaotic systems. Figure 20 shows the designed system. It shows that the current and generated voltage are converted from floating-point numbers to fixed-point numbers. As a result, these signals can be displayed in the oscilloscope.

### 6.2. FPGA Implementation Results

The designed discrete memristor and memristive chaotic maps are realized in Altera DE2-115 with EP4CE115F29C7. It is shown in Figure 21 that the physical implementation platform for the discrete memristor systems is carried out. The output of the Altera DE2-115 contains two 16-bit current and voltage signals, which are converted using the DAC8552 chip. Finally, the obtained results are displayed in the oscilloscope.

In the discrete memristor IP core, we set a=0.667 and $b=1.4828\times 10^{-4}$. The pinched hysteresis loops of the integer-order discrete memristor are presented in Figure 22. Meanwhile, the pinched hysteresis loops of the Caputo fractional-order discrete memristor are presented in Figure 23, and the pinched hysteresis loops of the G–L fractional-order discrete memristor are presented in Figure 24. They show that the pinched hysteresis loops displayed in the oscilloscope agree well with the corresponding Matlab simulation results and indicate that the designed memristor digital circuit can be used in the engineering application field.

Meanwhile, the discrete memristive chaotic maps are realized in the FPGA digital circuit. Let ε=3 and *b* take values including $b=1.4828\times 10^{-4}$ and b=1.4828, the results are shown in Figure 25. This shows that the digital circuit implementation results agree well with the corresponding Matlab simulation results. The proposed fractional-order discrete memristive Sine map can be realized in the hardware circuit for further applications.

## 7. Conclusions

In this paper, we discussed the mathematical models for discrete memristors based on Caputo-like difference and G–L difference. Numerical simulations show that both kinds of discrete memristor satisfy the three fingerprints of a memristor device. The integer-order memristor can be deduced from both fractional-order memristors, although they are built through different fractional-order differences. Compared with the Caputo difference-based memristor, the G–L difference-based memristor works in a much lower frequency domain. The analysis results show that the integer-order memristor has the perfect memory effect, and the fractional-order memristor does not. The solution of the fractional-order memristor has a series of coefficients related to the fractional-order difference. Since the fractional-order discrete memristor is related all the historical data, to improve computational efficiency and to realize the discrete memristor in a digital circuit, discrete memristors with a short-term memory effect are built and analyzed. As a result, the characteristics of discrete memristors have some differences with the continuous memristors. Thus, it is a new way to model memristors, which can be used in discrete systems.

By introducing discrete memristors to the Sine map, three memristive chaotic maps are obtained. This shows that the system has higher complexity than the original Sine map. Moreover, FPGA digital circuit implementation of the fractional-order discrete memristors and chaotic maps are carried out. This shows that the proposed memristive systems can be realized in digital circuits. Finally, applications of discrete memristors and memristive chaotic systems will comprise our future work.

## Figures and Tables

**Figure 1 entropy-24-00786-f001:**
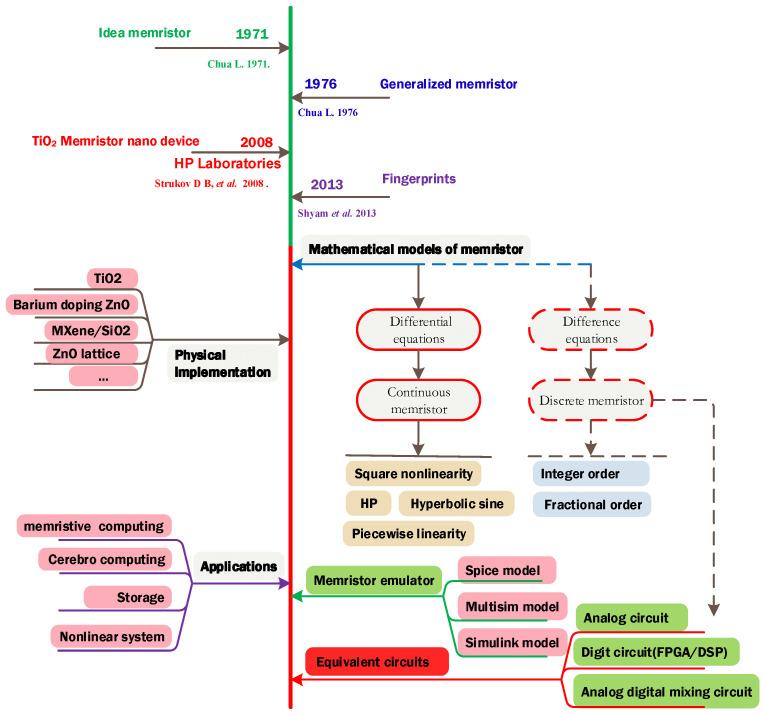
A short history of research about memristor and the position of this paper [1,2,3,41].

**Figure 2 entropy-24-00786-f002:**
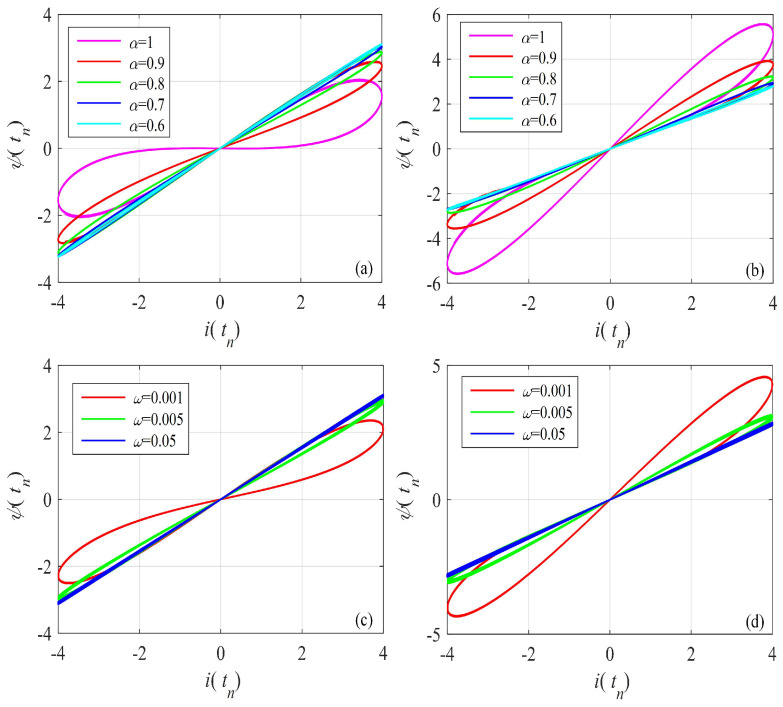
Pinched hysteresis loops of the G–L difference-based discrete memristor with $x(t_{0})$ = 0.1. (**a**) $g\left(x\right)=a_{1}+b_{1}x$, amplitude A=4 and frequency ω=0.001, and different derivative order; (**b**) $g\left(x\right)=a_{2}+b_{2}\left|x\right|$, amplitude A=4 and frequency ω=0.001, and different derivative order; (**c**) $g\left(x\right)=a_{1}+b_{1}x$, A=4, α=0.95 and different frequency ω; (**d**) $g\left(x\right)=a_{2}+b_{2}\left|x\right|$, A=4, α=0.95 and different frequency ω.

**Figure 3 entropy-24-00786-f003:**
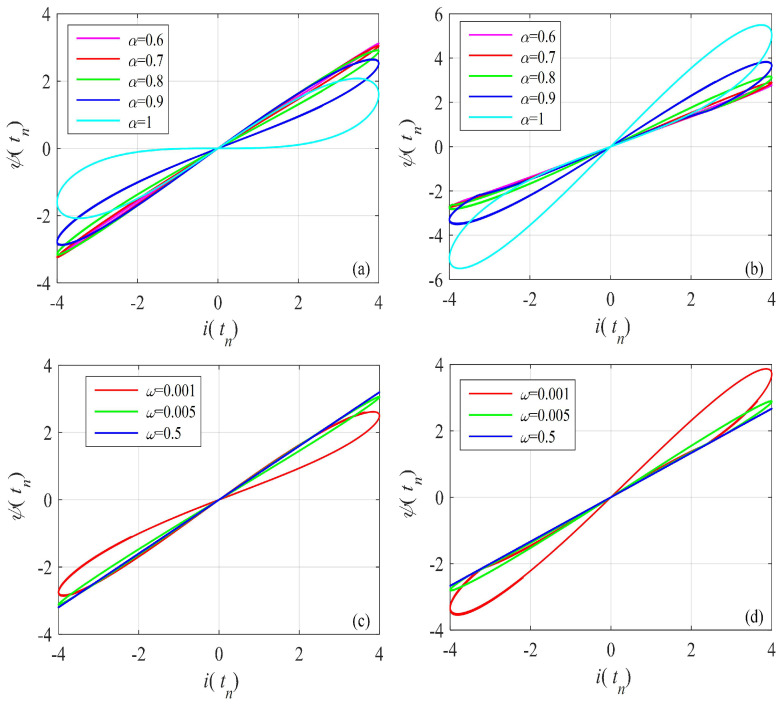
Pinched hysteresis loops of the Caputo-like difference-based discrete memristor with $x(t_{0})$ = 0.1. (**a**) $g\left(x\right)=a_{1}+b_{1}x$, amplitude A=4 and frequency ω=0.001, and different derivative order; (**b**) $g\left(x\right)=a_{2}+b_{2}\left|x\right|$, amplitude A=4 and frequency ω=0.001, and different derivative order; (**c**) $g\left(x\right)=a_{1}+b_{1}x$, A=4, α=0.9 and different frequency ω; (**d**) $g\left(x\right)=a_{2}+b_{2}\left|x\right|$, A=4, α=0.9 and different frequency ω.

**Figure 4 entropy-24-00786-f004:**
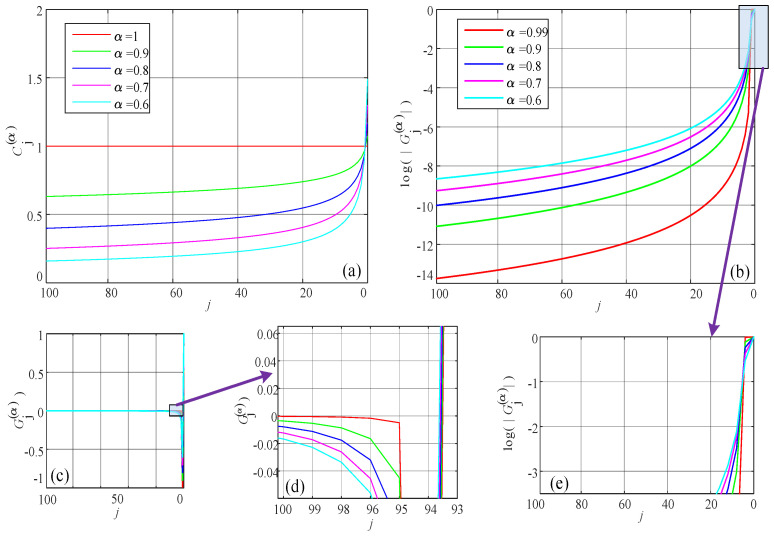
Memory effect of the fractional-order differences. (**a**) Caputo difference; (**b**–**e**) G–L difference, where (**e**) is the partial enlarged drawing of (**b**), and vertical coordinates of (**b**,**e**) are logarithmic, (**d**) is the partial enlarged drawing of (**c**).

**Figure 5 entropy-24-00786-f005:**
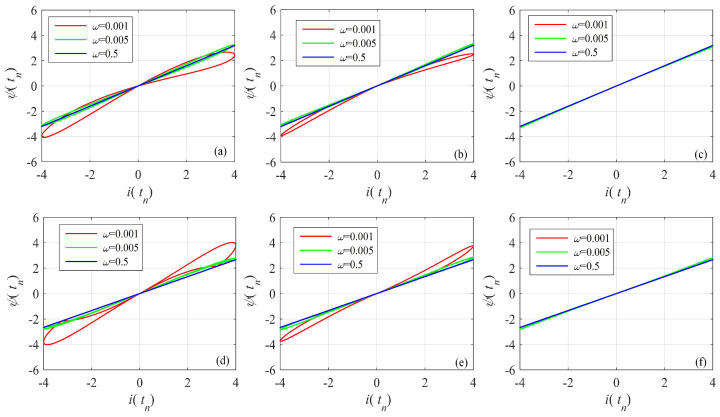
Pinched hysteresis loops of the Caputo difference-based discrete memristor with short-term memory effect. (**a**) $g\left(x\right)=a_{1}+b_{1}x$, and $L_{C}=2000$; (**b**) $g\left(x\right)=a_{1}+b_{1}x$, and $L_{C}=1000$; (**c**) $g\left(x\right)=a_{1}+b_{1}x$, and $L_{C}=100$; (**d**) $g\left(x\right)=a_{2}+b_{2}\left|x\right|$, and $L_{C}=2000$; (**e**) $g\left(x\right)=a_{2}+b_{2}\left|x\right|$, and $L_{C}=1000$; (**f**) $g\left(x\right)=a_{2}+b_{2}\left|x\right|$, and $L_{C}=100$.

**Figure 6 entropy-24-00786-f006:**
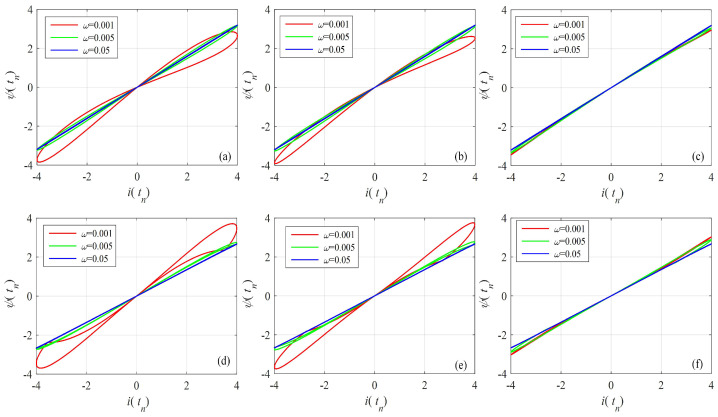
Pinched hysteresis loops of the G–L difference-based discrete memristor with short-term memory effect. (**a**) $g\left(x\right)=a_{1}+b_{1}x$, and $L_{GL}=100$; (**b**) $g\left(x\right)=a_{1}+b_{1}x$, and $L_{GL}=50$; (**c**) $g\left(x\right)=a_{1}+b_{1}x$, and $L_{GL}=10$; (**d**) $g\left(x\right)=a_{2}+b_{2}\left|x\right|$, and $L_{GL}=100$; (**e**) $g\left(x\right)=a_{2}+b_{2}\left|x\right|$, and $L_{GL}=50$; (**f**) $g\left(x\right)=a_{2}+b_{2}\left|x\right|$, and $L_{GL}=10$.

**Figure 7 entropy-24-00786-f007:**
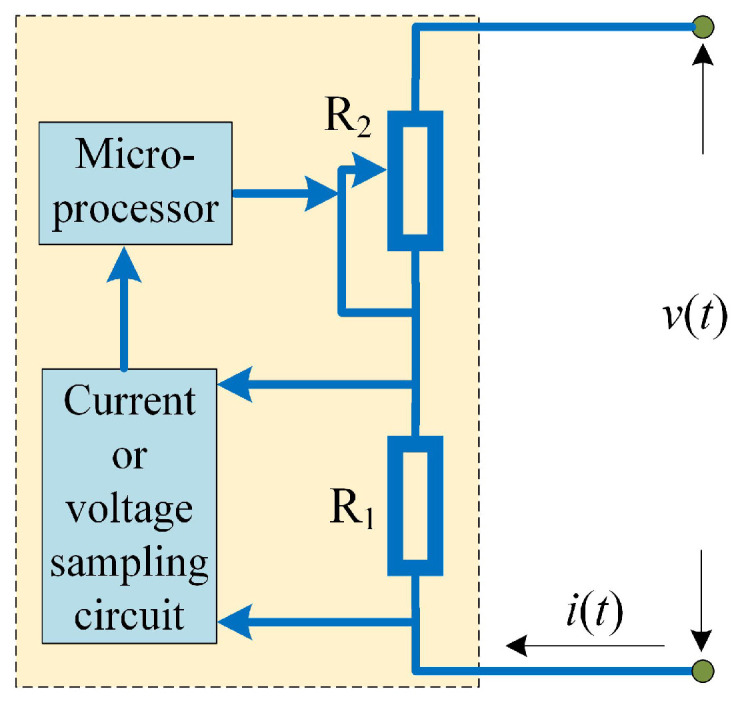
A physical realization scheme of the discrete memristor for analog and digital circuits (Reproduced with permission from [47] [Guangyi Wang], [J. Hangzhou Dianzi Univ. Nat. Sci]; published by [J. Hangzhou Dianzi Univ. (Natural Science edition)], [2018]).

**Figure 8 entropy-24-00786-f008:**
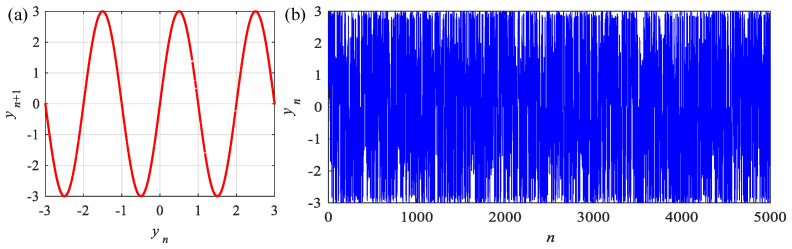
Phase diagram and time series of the Sine map. (**a**) Phase diagram; (**b**) Time series.

**Figure 9 entropy-24-00786-f009:**
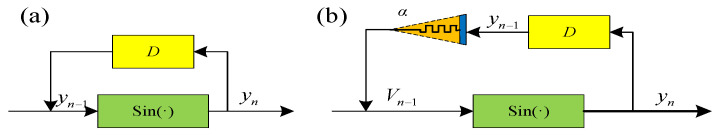
The block diagrams for the (**a**) chaotic map and (**b**) chaotic map with discrete memristor.

**Figure 10 entropy-24-00786-f010:**
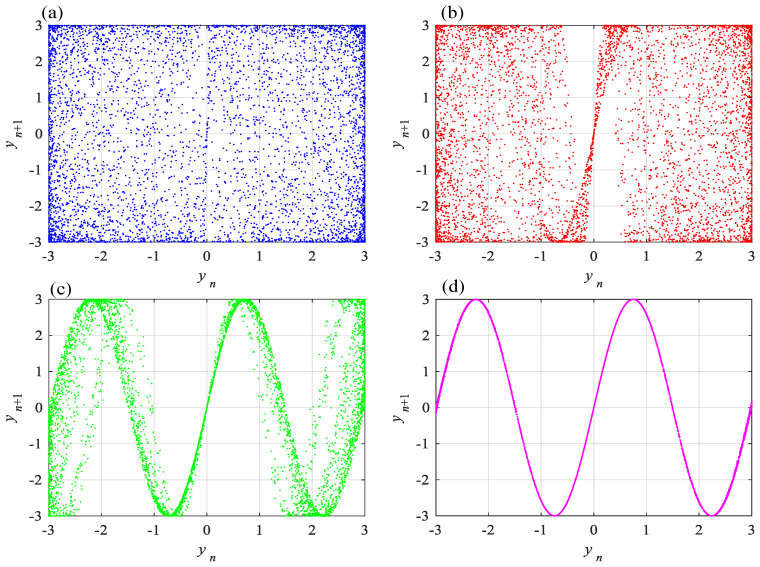
Phase diagrams of the Caputo fractional-order chaotic memristive Sine map with different parameter *b*. (**a**) $b=1.4828\times 10^{-1}$; (**b**) $b=1.4828\times 10^{-2}$; (**c**) $b=1.4828\times 10^{-3}$; (**d**) $b=1.4828\times 10^{-4}$.

**Figure 11 entropy-24-00786-f011:**
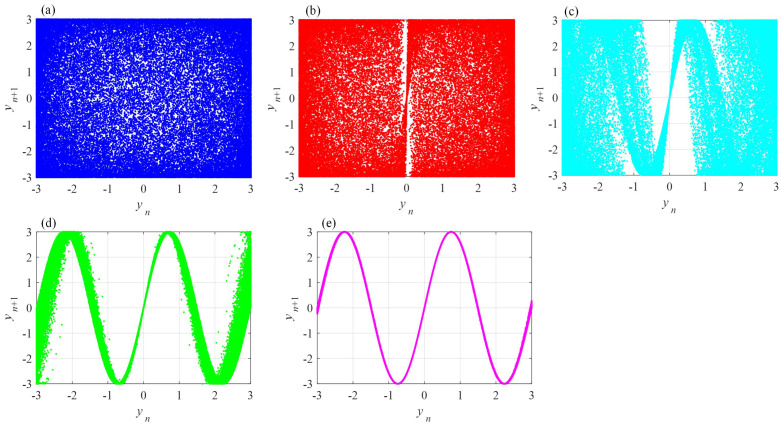
Phase diagrams of fractional-order chaotic memristive Sine map with different parameter *b*. (**a**) b=1.4828; (**b**) $b=1.4828\times 10^{-1}$; (**c**) $b=1.4828\times 10^{-2}$; (**d**) $b=1.4828\times 10^{-3}$; (**e**) $b=1.4828\times 10^{-4}$.

**Figure 12 entropy-24-00786-f012:**
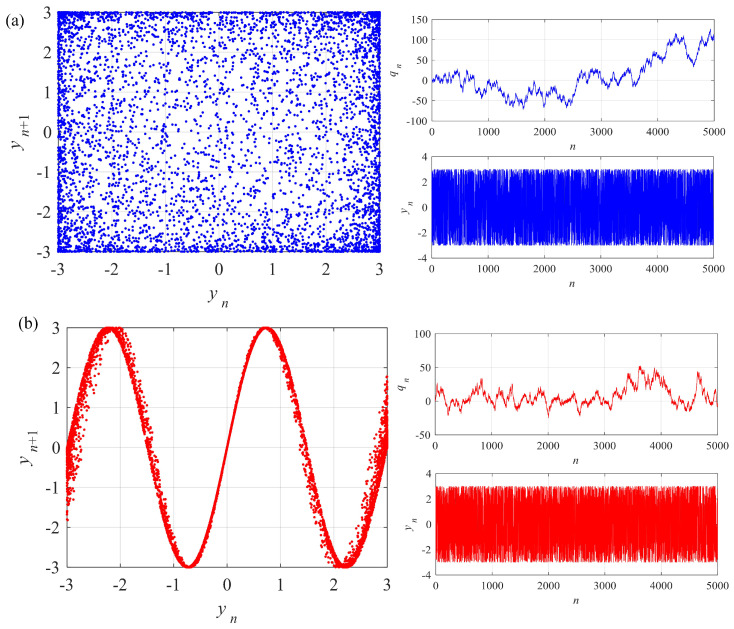
Phase diagrams and time series of the integer-order memristive Sine map with different parameter *b*. (**a**) b=1.4828; (**b**) $b=1.4828\times 10^{-3}$.

**Figure 13 entropy-24-00786-f013:**
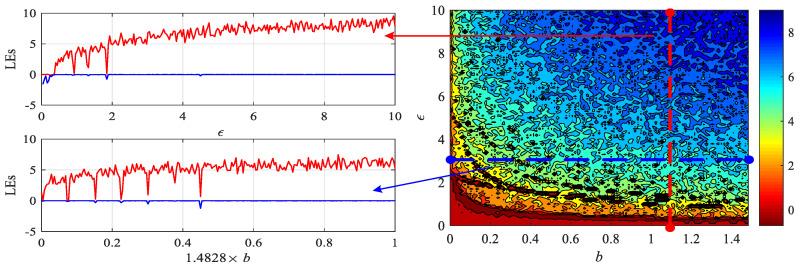
Maximum LEs of the integer-order discrete memristive Sine map.

**Figure 14 entropy-24-00786-f014:**
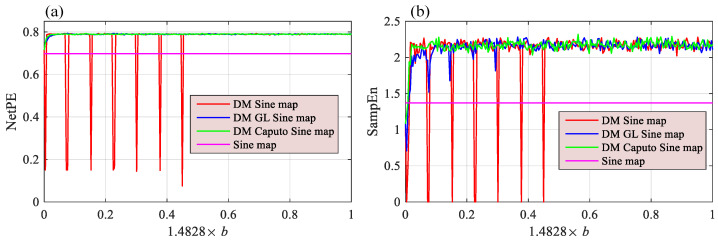
Complexity analysis results of the Sine maps with *b* varying measured by different methods (**a**) NetPE; (**b**) SampEn.

**Figure 15 entropy-24-00786-f015:**
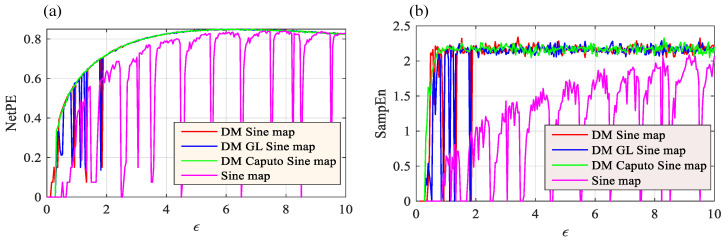
Complexity analysis results of the Sine maps with ε varying measured by different methods (**a**) NetPE; (**b**) SampEn.

**Figure 16 entropy-24-00786-f016:**
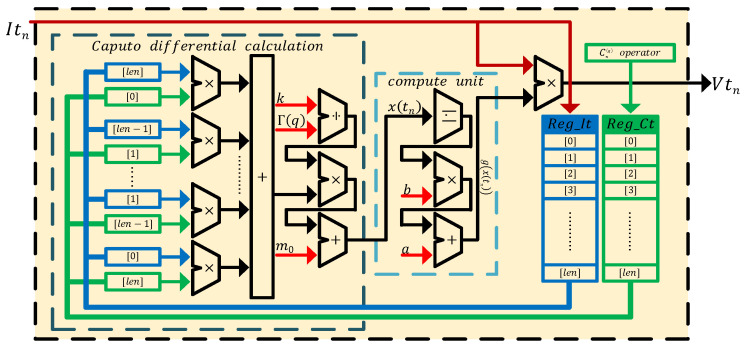
Digital circuit of the Caputo fractional-order discrete memristor.

**Figure 17 entropy-24-00786-f017:**
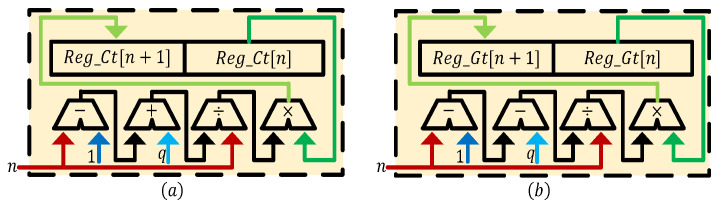
(**a**) Digital circuit of the $C_{n}^{q}$ operator module; (**b**) Digital circuit of the $G_{n}^{q}$ operator module.

**Figure 18 entropy-24-00786-f018:**
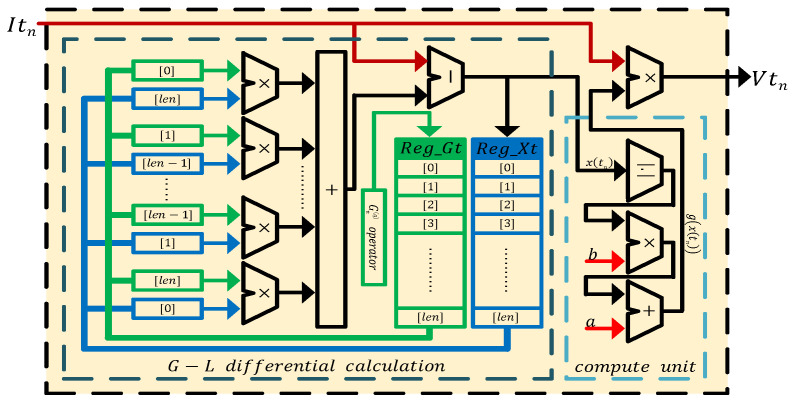
Digital circuit of the G–L fractional-order discrete memristor.

**Figure 19 entropy-24-00786-f019:**
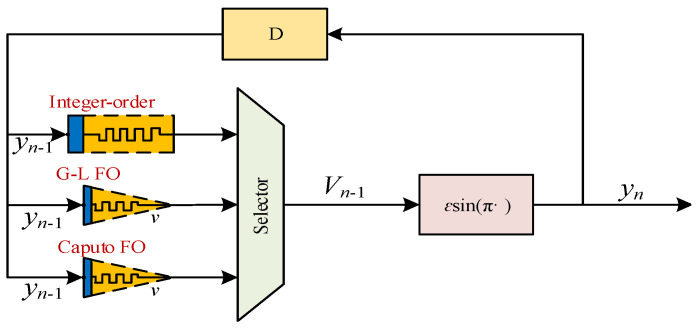
The data path circuit of the discrete memristive Sine map.

**Figure 20 entropy-24-00786-f020:**
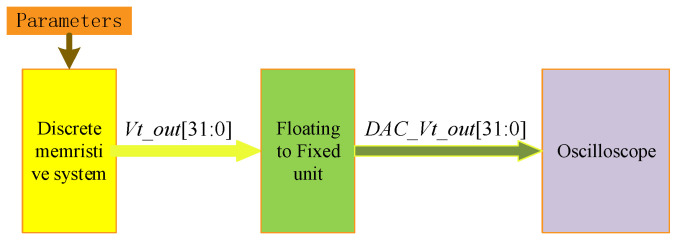
The designed system test circuit.

**Figure 21 entropy-24-00786-f021:**
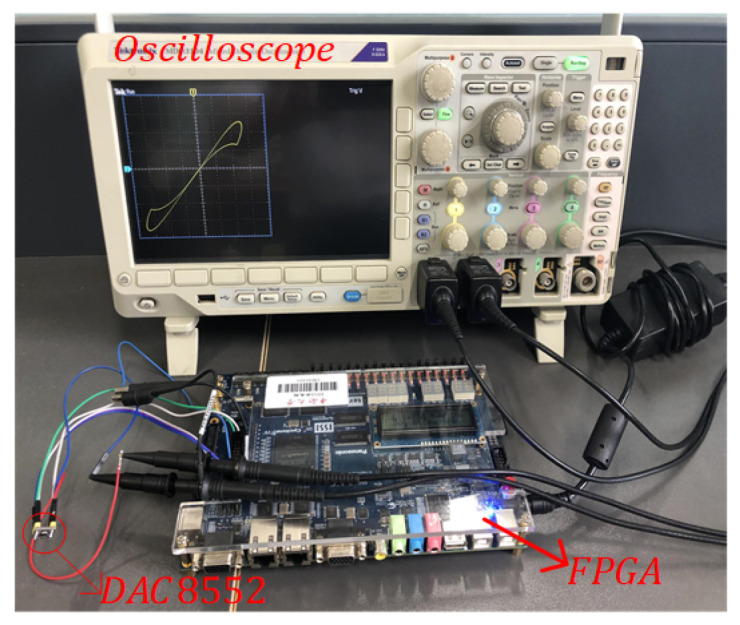
The physical implementation platform of the discrete memristive system.

**Figure 22 entropy-24-00786-f022:**
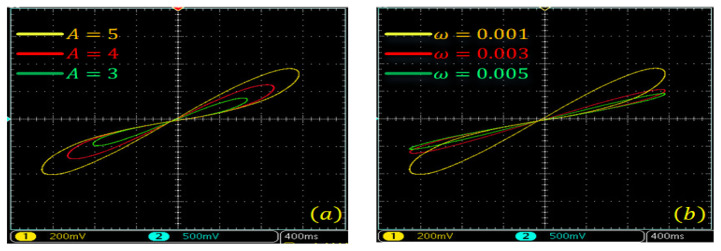
Pinched hysteresis loops of the integer-order discrete memristor. (**a**) Different *A*; (**b**) Different ω.

**Figure 23 entropy-24-00786-f023:**
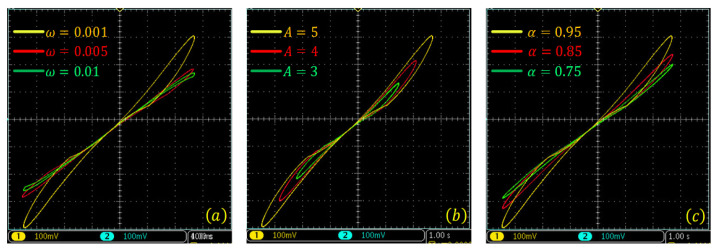
Pinched hysteresis loops of the Caputo-like difference-based-discrete memristor. (**a**) Different ω; (**b**) Different *A*; (**c**) Different α.

**Figure 24 entropy-24-00786-f024:**
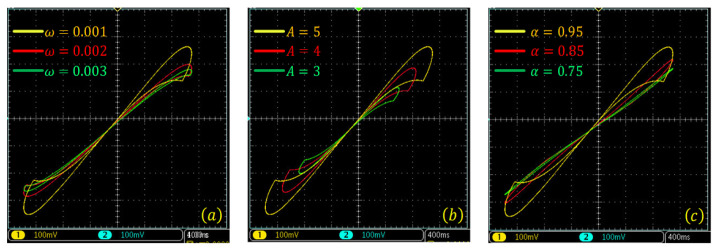
Pinched hysteresis loops of the G–L difference-based-discrete memristor. (**a**) Different ω; (**b**) Different *A*; (**c**) Different α.

**Figure 25 entropy-24-00786-f025:**
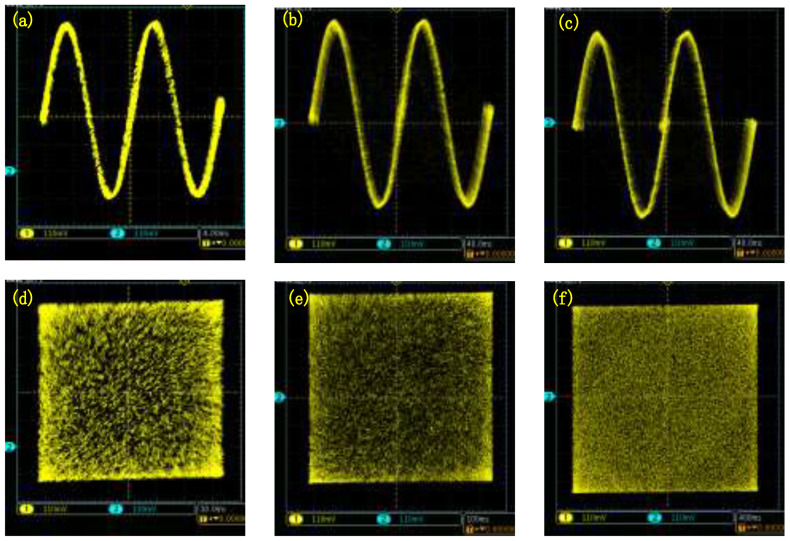
Phase diagrams of the chaotic maps from the FPGA digital circuit. (**a**) Integer-order system with $b=1.4828\times 10^{-4}$; (**b**) G–L fractional-order system with $b=1.4828\times 10^{-4}$; (**c**) Caputo fractional-order system with $b=1.4828\times 10^{-4}$; (**d**) Integer-order system with b=1.4828; (**e**) G–L fractional-order system with b=1.4828; (**f**) Caputo fractional-order system with b=1.4828.

**Table 1 entropy-24-00786-t001:** Types, formulas and symbols of generalized memristors.

Memristor	Integer-Order		Fractional-Order	
	Formula	Symbol	Formula	Symbol
Charge-controlled memristor	$\begin{array}{c} V=M\left(q,i\right)i\\ \frac{dq}{dt}=ki \end{array}$	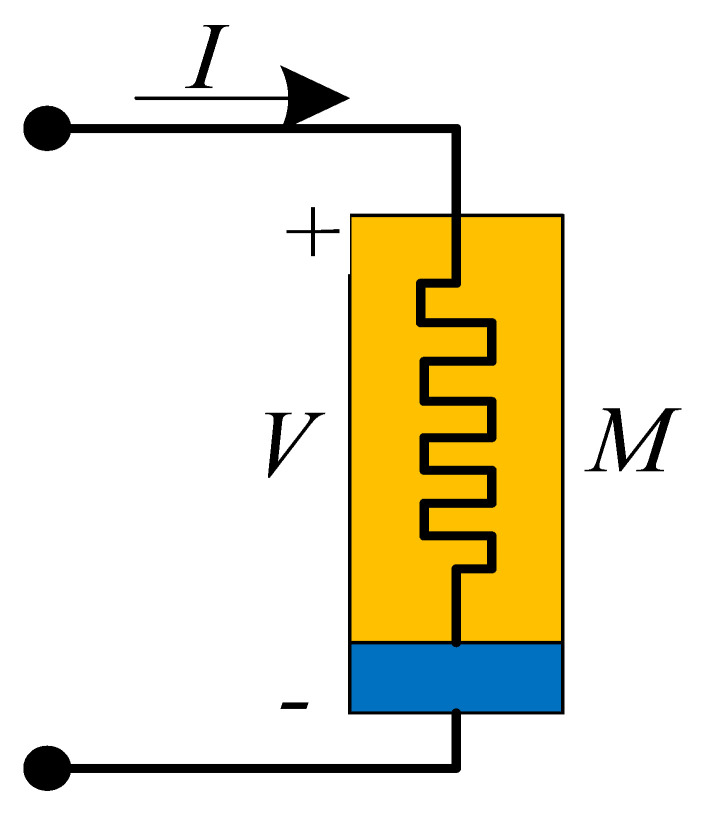	$\begin{array}{c} V=M\left(q,i\right)i\\ D_{t_{0}}^{\alpha }q=ki \end{array}$	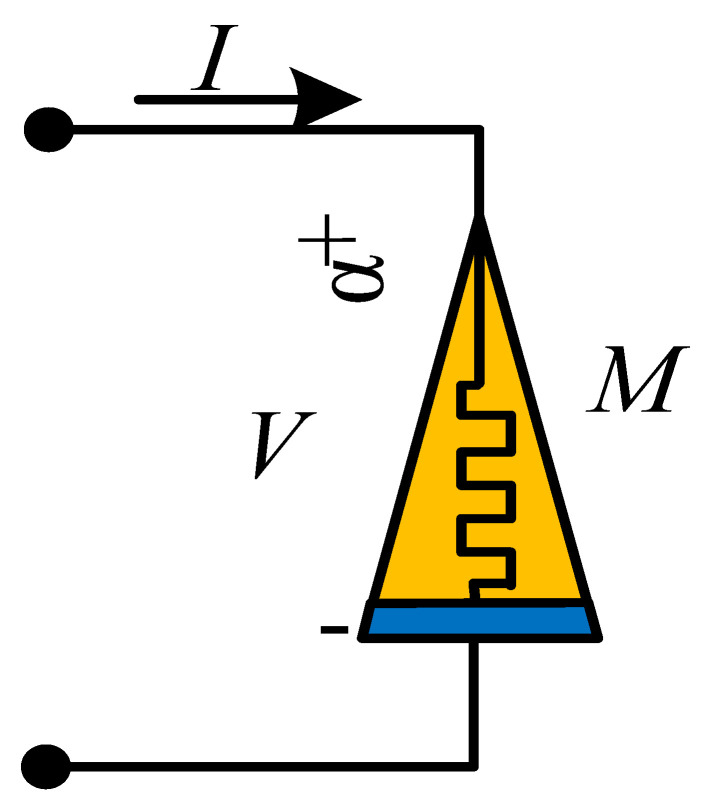
Magnetron controlled memristor	$\begin{array}{c} i=G\left(\varphi ,V\right)V\\ \frac{d\varphi }{dt}=kV \end{array}$	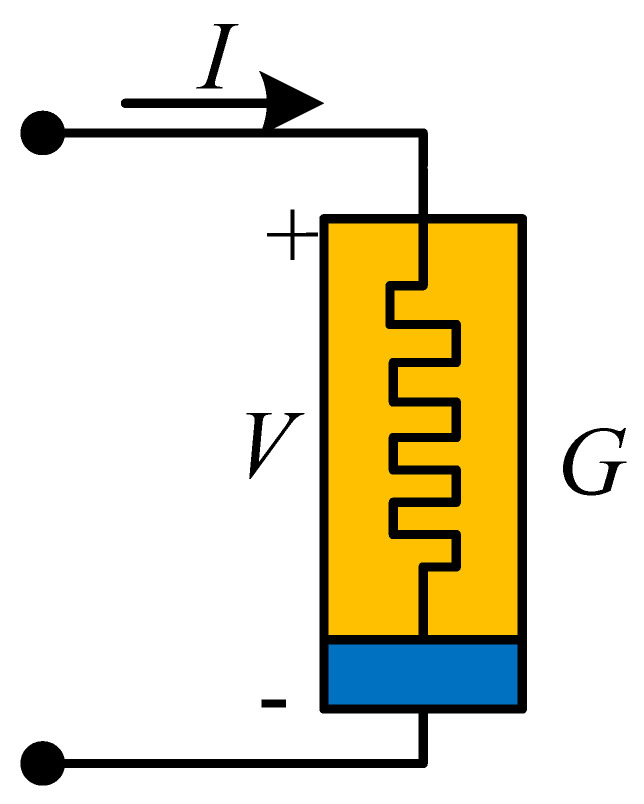	$\begin{array}{c} i=G\left(\varphi ,V\right)V\\ D_{t_{0}}^{\alpha }\varphi =kV \end{array}$	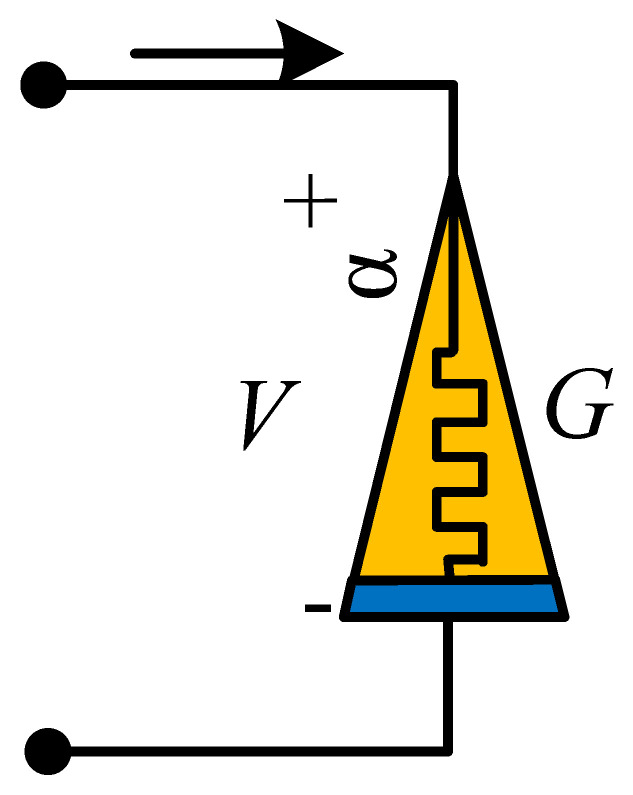
Discrete memristor	$\begin{array}{c} y_{out}=F\left(\varphi ,x_{in}\right)x_{in}\\ \Delta \varphi =f\left(\varphi ,x_{in}\right) \end{array}$	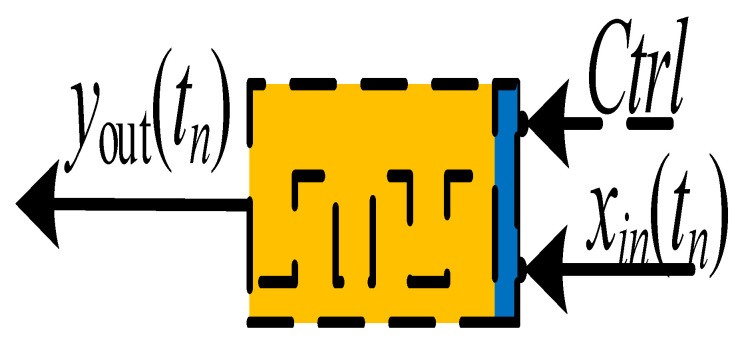	$\begin{array}{c} y_{out}=F\left(\varphi ,x_{in}\right)x_{in}\\ \Delta^{\alpha }\varphi =f\left(\varphi ,x_{in}\right) \end{array}$	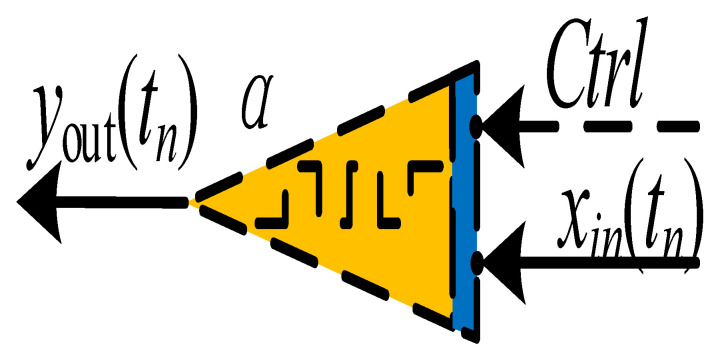

**Table 2 entropy-24-00786-t002:** The characters of different kinds of memristors.

Type	Characteristics	Applications	Modeling
Continuous memristor	🟉 Memory effect; 🟉 Mathematical models are mature; 🟉 Nano-device implementation; 🟉 The parameters of memristor are fixed.	🟉 Continuous system; 🟉 Differential system	🟉 Differential operator; 🟉 Analog circuit
Discrete memristor	🟉 Memory effect 🟉 Mathematical models is not mature 🟉 Nano-device implementation 🟉 Digital circuit realization of controllability 🟉 Input and output can be digital sequences	🟉 Discrete system; 🟉 Difference system; 🟉 Iterative system; 🟉 Signal processing	🟉 Difference operator; 🟉 Digital circuit; 🟉 Mixed circuit

## Data Availability

All data and models during the study appear in the submitted article.

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
