# Peer review of "Discrete Memristor and Discrete Memristive Systems"

_entropy, 2022, doi:10.3390/e24060786_

Round 1

Reviewer 1 Report

This work is well developed and the results obtained experimentally agree with the proposed models of discrete memristor and discrete memrisistive system. I believe that the authors can improve the understanding of the article for how I specified it in the attached file observations.

Reviewer 2 Report

This long paper presents models and numerical analyses for discrete memristor, and characterizes some of their properties. The work generalizes previous work on this kind of devices and provides insight into their main properties and applications, including memory and chaotic behavior. The paper is quite long and dense, and additionally needs some corrections in the following issues:

1) Definitions 2 and 4 are actually propositions or theorems. Actually, Definition 4 is followed by a proof.

2) The mathematical models for the fractional difference systems depend heavily on the choice of the control functions f(·) and g(·), but examples of these are deferred until the numerical study. The authors should better motivate why they choose those particular functions.

3) The complexity analysis (Sect. 5.5.2), summarized in figures 14 and 15 are difficult to follow and interpret. Please clarify the meaning of complexity and the physical significance of the complexity measures.

4) There are frequent typos and variations in the nomenclature: memrisive, memerisive, memrisistive... Please, unify and consolidate the names.

Reviewer 3 Report

The article contains a fairly complete and well-structured review of mathematical models used in the analysis of various types of memristive systems. The presented approaches to the description of discrete memristive systems are original and may be of interest to specialists. The article corresponds to the subject of the journal and can be published in the presented form.

Round 2

Reviewer 2 Report

The authors have addressed properly and satisfactorily all my previous concerns. While the paper is long, it contains enough contribution and interest to be accepted for publication.

Author Response

Please find the attached  response in the attahced file. 
